

# Contrasting responses of vegetation productivity to intraseasonal rainfall in Earth System Models

Bethan L. Harris[1,2], Tristan Quaife[2, 3], Christopher M. Taylor[1, 2], and Phil P. Harris[1, 2]

[1]UK Centre for Ecology & Hydrology, Wallingford, UK
[2]National Centre for Earth Observation, UK
[3]University of Reading, Reading, UK

**Correspondence:** Bethan L. Harris (bethar@ceh.ac.uk)

**Abstract.** Correctly representing the response of vegetation productivity to water availability in Earth System Models (ESMs) is essential for accurately modelling the terrestrial carbon cycle and the evolution of the climate system. We evaluate this response at the intraseasonal timescale in five CMIP6 ESMs by analysing changes in Gross Primary Productivity (GPP) after intraseasonal rainfall events and comparing to the responses found in a range of observation-based products. When composited

around all intraseasonal rainfall events globally, both the amplitude and the timing of the GPP response show large inter-model differences, demonstrating discrepancies between models in their representation of water-carbon coupling processes. However, the responses calculated from the observational datasets also vary considerably, making it challenging to assess the realism of the modelled GPP responses. The models correctly capture that larger increases in GPP at the regional scale are associated with larger increases in surface soil moisture and larger decreases in atmospheric vapour pressure deficit. However, the sensitivity

of the GPP response to these drivers varies between models. The GPP in NorESM is insufficiently sensitive to surface soil moisture perturbations when compared to any observational GPP product tested. Most models produce a faster GPP response where the surface soil moisture perturbation is larger, but the observational evidence for this relationship is weak. This work demonstrates the need for a better understanding of the uncertainties in the representation of water-vegetation relationships in ESMs, and highlights a requirement for future daily-resolution observations of GPP to provide a tighter constraint on global

water-carbon coupling processes.

## 1   Introduction

The flux of carbon into the land surface resulting from photosynthesis is referred to as Gross Primary Productivity (GPP). Terrestrial GPP is the single largest component flux in the global carbon cycle, with typical estimates of its magnitude being

in the region of 140 PgCa$^{-1}$ (Canadell et al., 2021). As such, it plays a key role in modulating the atmospheric carbon dioxide concentration and the terrestrial biosphere has taken up around a third of anthropogenic $CO_2$ emissions over recent decades (Friedlingstein et al., 2022). Understanding the processes that control this uptake is critical to our ability to correctly



model climate because any change has potentially significant implications for the accumulation of $CO_2$ in the atmosphere. Relevant processes include a number of limiting factors, such as the availability of nutrients (Fernández-Martínez et al., 2014),
fertilisation by atmospheric carbon dioxide (Chen et al., 2022) and water— the focus of this study—both in terms of soil moisture that is accessible to plants and the atmospheric vapour pressure deficit.

Without sufficient available water for transpiration, or when there is high atmospheric demand (corresponding to a high vapour pressure deficit, VPD), plants close their stomata. As a consequence, they are unable to replenish leaf internal carbon dioxide, and thus photosynthesis is down–regulated. The water and carbon cycles are thus intrinsically linked, and the avail-
ability of water is a first order determinant of terrestrial GPP. There is growing evidence that the growth rate of atmospheric $CO_2$ is sensitive to changes in terrestrial water storage (Humphrey et al., 2018; Liu et al., 2023) and a reasonable hypothesis is that this signal is in part controlled by the response of photosynthesis to water availability in soil. In climate models, the representation of down–regulation of GPP due to lack of available soil water is typically quite crude. A commonly used approach is to model the potential, un-stressed GPP, and then apply a scalar which is determined by a linear ramp between some
bounds, below which no photosynthesis occurs, and above which it is unaffected by soil moisture. This avoids the complexities of implementing more process-based schemes, such as that of Bonan et al. (2014), into climate models, as these introduce additional poorly-constrained parameterisations. However, the schemes currently employed are sensitive to choices such as soil hydraulic parameters and the depth over which the soil moisture stress factor is calculated (Harper et al., 2021). These decisions, implemented by modelling groups, will lead to differences in the response of the modelled GPP to soil moisture.

It is generally the case that the stomatal conductance decreases with increasing VPD, and hence lowers the rate at which $CO_2$ is drawn into the substomatal cavity to replenish that used in photosynthesis. However, empirical studies reveal a complex relationship between water use efficiency (WUE, the amount of carbon assimilation per unit of water transpired) and VPD, and consequently, the relationship with GPP is not straightforward (Grossiord et al., 2020). The land surface components of many climate models use variants of the Ball-Berry scheme (Ball et al., 1987). This approach relates the stomatal conductance
to the carbon assimilation and atmospheric humidity via empirical parameters, assuming a linear response. Commonly used extensions to Ball-Berry in climate models, in particular that developed by Leuning (1995), provide more realistic responses of stomatal conductance to VPD but still rely on empirical parameterisation and do not appear to infer much additional skill over the original Ball-Berry model when used for large scale simulations (Knauer et al., 2015). As a consequence the relationship between GPP and VPD in climate models remains fairly crude. A key assumption in many climate models is that the slope of
the relationship between stomatal conductance and VPD can be assigned globally (often on a per-PFT basis) and is not affected by environmental conditions (Knauer et al., 2015). A further confounding factor is that the interpretation of the effects of VPD and soil moisture on GPP tend to be complicated by their interdependence on each other. High VPD will generally coincide with dry soils and vice-versa, due to feedbacks between surface energy and water fluxes and the lower atmosphere.

A key problem in trying to evaluate the processes in climate models involved in the water controls on photosynthesis in
models is that observations of GPP on large spatial scales tend to be heavily modelled themselves. For example the MODIS GPP product (MOD17) is based on a light use efficiency approach where the primary satellite data input is the fraction of absorbed photosynthetically active radiation (Running et al., 2004). The efficiency by which the absorbed light is used to drive





carbon assimilation is modelled, with meteorological drivers as inputs. Consequently, any difference between GPP predicted by a climate model and the corresponding MODIS data could be due to the modelled efficiency term in MOD17. Arguably,

a better point of contact between the models and the MODIS data would be the fAPAR itself as this is more directly derived from the actual satellite observations (Zobitz et al., 2014), but this contains no direct information about water.

Another approach to estimating GPP on global scales, which has been gaining much traction in recent years, is to upscale eddy covariance based estimates of GPP from flux tower networks, using EO and meteorological data as inputs to regression models. The FLUXCOM GPP product, for example, uses a wide range of data from the MODIS sensors, including vegetation

indices and land cover data, meteorological data from ERA-Interim, and a range of machine learning techniques to upscale the FLUXNET data record (Jung et al., 2020a; Tramontana et al., 2016a). Whilst these methods show great promise, their reliance on the flux towers means that the data they are trained on contains sampling bias toward northern temperate regions, with relatively little data across the tropics. This is particularly problematic as the majority of global GPP occurs in these tropical regions, so this data scarcity is a key contributor to uncertainty in global total GPP (Schimel et al., 2015). The tropics are also

likely to be a key region in which water will influence vegetation productivity as the climate changes (Worden et al., 2021).

Solar induced fluorescence (SIF) observations from space offer a new and potentially transformative information source on large scale GPP (Pickering et al., 2022). SIF is a by-product of photosynthesis, and observations are hence directly related to the bio-chemical mechanisms controlling GPP in a way not available from other global scale data sources. However, mechanistic estimates of GPP from satellite observed SIF are not yet routinely available, and most existing products are generated using

statistical regression (Bai et al., 2022; Alemohammad et al., 2017). In addition, the EO-based SIF data record is relatively short, with longer term data records (such as from GOSAT) having relatively sparse spatial sampling.

Previous evaluations of GPP in global models have typically focused on annual mean GPP, interannual variability and trends (Piao et al., 2013; Anav et al., 2015; Slevin et al., 2017; Kim et al., 2018; Hu et al., 2022). These studies have shown that although models can produce reasonable spatial and seasonal distributions of GPP, there is a large inter-model range in mean

global GPP (Piao et al., 2013; Anav et al., 2015; Kim et al., 2018; Hu et al., 2022). The observation-based products used for evaluation also have large differences in mean global GPP (Anav et al., 2015). Models produce a mean annual GPP that is sensitive to interannual variations in temperature, precipitation and radiation (Piao et al., 2013; Anav et al., 2015), but there is uncertainty in the magnitude of the GPP interannual variability, and it tends to be too weak in observation-based products compared to models.

However, studies using daily observational data have shown that vegetation also responds to intraseasonal variability in precipitation (Guan et al., 2014; Wu et al., 2022; Harris et al., 2022). Evaluating modelled GPP responses at these shorter timescales offers an opportunity to investigate the physical processes linking climate drivers to GPP. In many regions, the intraseasonal variability of vegetation strongly influences vegetation-climate coupling out to interannual timescales (Guan et al., 2014, 2018; Wu et al., 2021; Barnes et al., 2021). Correctly modelling GPP responses to shorter-term variations in

precipitation could therefore improve estimates of land carbon uptake on climate-relevant timescales.

Here, we investigate how GPP (and other linked variables, such as soil moisture and vapour pressure deficit) responds to intraseasonal precipitation events in different CMIP6 models and assess against a range of observation-based GPP products to





get physical insight into which models have realistic rainfall–GPP coupling processes. An issue in comparing climate models outputs in terms of their rainfall–GPP responses is that the timing, magnitude and location of precipitation events is likely to be inconsistent. Consequently any differences in the modelled GPP could be due to differences in the simulation of the rainfall regime itself. The approach we have adopted is to identify the timing of peak intraseasonal precipitation events (Harris et al., 2022) and compare the response of other variables relative to that point in time. We show that, whilst models agree on the timescales of the response of soil moisture to precipitation events, there are considerable discrepancies between models in the response of GPP to precipitation. This points to disagreement in the processes that couple available water and GPP in the models. Furthermore, we note that observational GPP products also exhibit similar levels of disagreement in the timing of response of GPP to rainfall, pointing to the need for better data products.

## 2 Data and Methods

### 2.1 CMIP6 model data

We compare vegetation productivity responses in models from the CMIP6 *esm-hist* experiment (Eyring et al., 2016). This experiment covers the recent historical period from 1850–2014. Only data from 2000–2014 is included in our analysis, in order to permit a fair comparison with the available satellite observations. To investigate vegetation productivity responses to intraseasonal rainfall variability, we require daily or sub-daily model output data for precipitation and GPP. These data are available for the *esm-hist* experiment for only five models: ACCESS-ESM1-5, BCC-CSM2-MR, CNRM-ESM2-1, NorESM2-LM, and UKESM1-0-LL. We use the precipitation fields provided at daily resolution, and take the daily mean of the GPP data, which is made available at 3-hourly resolution. These models also all provide daily surface soil moisture (*mrsos*) data: this is the mass of water in the upper 10cm of soil. We compute daily near-surface VPD from near-surface air temperature (*tas*) and relative humidity (*hurs*). Near-surface relative humidity is not available at daily resolution for BCC-CSM2-MR, so this model is excluded from the VPD-focused sections of our analysis. All model data are regridded to $1° \times 1°$ horizontal resolution using land-area-weighted averaging.

### 2.2 Observational datasets

Intraseasonal rainfall events are identified using the Integrated Multi-satellitE Retrievals for GPM (IMERG) V06 daily product (Huffman et al., 2019). Surface soil moisture is assessed using both the ESA CCI Soil Moisture combined active-passive microwave product v06.1 (Dorigo et al., 2017; Gruber et al., 2019) and the Global Land Evaporation Amsterdam Model (GLEAM) v3.6a (Miralles et al., 2011; Martens et al., 2017). GLEAM models the distribution of soil water content based on Multi-Source Weighted-Ensemble Precipitation (MSWEP) v2.8 (Beck et al., 2017) and corrects the soil moisture of the top model layer by assimilating the ESA CCI Soil Moisture combined product. The two surface soil moisture datasets are therefore not independent, but both are tested here to ascertain whether the differences in the products affect the assessment of the CMIP6 models' surface soil moisture. ESA CCI Soil Moisture has the advantage of being less reliant on model algorithms,





while the top soil layer in GLEAM is 10cm deep and may therefore be more representative of the CMIP *mrsos* variable, which

is defined over the same depth. The microwave observations used in ESA CCI Soil Moisture measure varying soil depths, but are typically taken to quantify soil moisture in the top 2–5cm (Ulaby et al., 1982). We compute near-surface VPD from ERA5 reanalysis data for 2m air temperature and 2m dewpoint temperature (Hersbach et al., 2020).

Daily GPP data is obtained from FLUXCOM RS+METEO (Jung et al., 2020b). FLUXCOM RS+METEO uses machine learning to upscale estimates of terrestrial carbon fluxes from eddy covariance flux towers to create a global gridded GPP

product based on remote sensing and meteorological forcing data. We test versions of FLUXCOM RS+METEO using two different meteorological forcing datasets: ERA5 and CRU JRA v1.1 (Harris et al., 2014; Kobayashi et al., 2015; Harris, 2019). For each of these forcing datasets, we use the ensemble median GPP over three machine learning methods and two flux partitioning methods, as detailed by Tramontana et al. (2016b) and Jung et al. (2019). Note that while the meteorological forcing data is updated daily, the remote sensing driving data are based on mean seasonal cycles (Jung et al., 2020b).

Whilst daily data is preferable for investigating intraseasonal variability, information can also be obtained from other sub-monthly GPP datasets. Incorporating these datasets into the analysis allows a better understanding of the uncertainty in our results that arises from the need to derive GPP from direct observations. We include the 8-daily MODIS Terra GPP product (Running et al., 2015), as quality controlled and regridded by Kern (2021). We also analyse the 8-daily VPM GPP product (Zhang et al., 2017a), which is intended to provide an alternative to MODIS Terra GPP by using the improved light use effi-

ciency theory of the vegetation photosynthesis model (VPM). The VODCA2GPP dataset (Wild et al., 2022), which estimates GPP based on Vegetation Optical Depth (VOD) retrieved from passive microwave observations, is also produced at 8-day resolution.

Satellite observations of solar-induced chlorophyll fluorescence (SIF) provide an additional way of estimating GPP at the global scale. SIF is approximately linearly correlated with GPP (Frankenberg et al., 2011), although the relationship varies

with season and vegetation type (Chen et al., 2021). We therefore also compare the modelled GPP responses with SIF data that is spatially downscaled from GOME-2 observations (Duveiller et al., 2020). This product includes data based on two different methods of retrieval from GOME-2: Joiner et al. (2013), henceforth labelled JJ, and Köhler et al. (2015), labelled PK. These datasets have a temporal resolution of 8 days, using a 16-day rolling window.

All observational products are available for the complete period 2000–2014, except for IMERG, which is available from

June 2000 onwards, and the GOME-2 SIF data, which is available from 2007 onwards. All products are regridded by land-area-weighted averaging to a horizontal resolution of 1°×1°.

## 2.3   Compositing around intraseasonal precipitation events

Intraseasonal precipitation events are identified using the method of Harris et al. (2022). Long-term linear temporal trends in the data are removed before processing. For each 1°×1° grid box, we apply a 25-day low-pass Lanczos filter to the daily

precipitation anomaly, where the anomaly is computed relative to the climatology for a 7-day rolling window. Precipitation events are then defined as local maxima of the filtered time series that lie above one standard deviation from the mean. The dates of these events are identified separately for each CMIP6 model and for IMERG. We then composite daily standardised





anomalies (relative to 7-day rolling climatology) of precipitation, SSM, VPD and GPP around the dates of the intraseasonal precipitation events. For the observational GPP products with 8-day resolution, the standardised anomalies are computed
relative to a 31-day rolling climatology instead of 7-day, to ensure sufficient data for the climatological averaging. It should be noted that the composited variables do not undergo Lanczos filtering; the filter is only used to determine the dates of the precipitation events around which to composite.

The results are aggregated over regions containing multiple grid boxes to increase the number of precipitation events contributing to each composite. Only grid boxes with valid ESA CCI Soil Moisture observations are included, to maximise our
observational knowledge of the surface soil moisture perturbations that are driving the GPP responses. Although this eliminates tropical forest regions from the analysis, it is found to make only a small difference to the resulting GPP composites, since vegetation in these regions does not respond strongly to intreaseasonal wet events compared to vegetation in water-limited regions (Harris et al., 2022). We also remove frozen grid boxes from the composites by discarding events in months when the median of maximum 2m air temperature is below 0°C.

## 3    Results

### 3.1    Global evaluation

Global (60°S–80°N) composites of precipitation, surface soil moisture and GPP around intraseasonal wet events are shown in Figure 1. The composites of precipitation are similar between the CMIP6 models and observations (Figure 1a), indicating that our method successfully creates similar standardised anomalies of precipitation driving the land surface responses in each case.
The surface soil moisture responses following these wet events are also similar between models (Figure 1b), with consistent maximum standardised anomalies 2–3 days after the wet event peak and comparable longer-term anomalies of elevated soil moisture out to 60 days. The observed surface soil moisture composites from the GLEAM and ESA CCI products are also a reasonable match to the CMIP6 models. The models are in better agreement with the observed composite obtained from GLEAM, which may be due to GLEAM representing the same depth of soil surface layer as the model output.
However, the responses of GPP to the intraseasonal wet events show large inter-model differences (Figure 1c). All models exhibit a positive GPP anomaly in the days following the peak of wet events, but the amplitude and timing of this anomaly varies greatly. CNRM-ESM and UKESM produce a larger standardised anomaly in GPP than the other models, while ACCESS-ESM and NorESM show the smallest responses. NorESM also has the slowest response in GPP, with the peak GPP anomaly occurring 22 days after the peak precipitation anomaly, compared to 5 days for BCC-CSM, the fastest-responding model. The
models also show contrasting behaviour leading up to the peak of the wet events: some have a much larger negative GPP anomaly than others, associated with a reduction in downwelling shortwave radiation during the wet events. However, in this study we focus on the post-event characteristics of the GPP response.

To investigate which of the modelled productivity responses are realistic, the wet event GPP composites are shown in Figure 1d for a variety of datasets based on observations (as described in Section 2). While composites obtained from the same family
of observational products (e.g. the two SIF retrieval methods) are similar to one another, there is a large spread in the responses



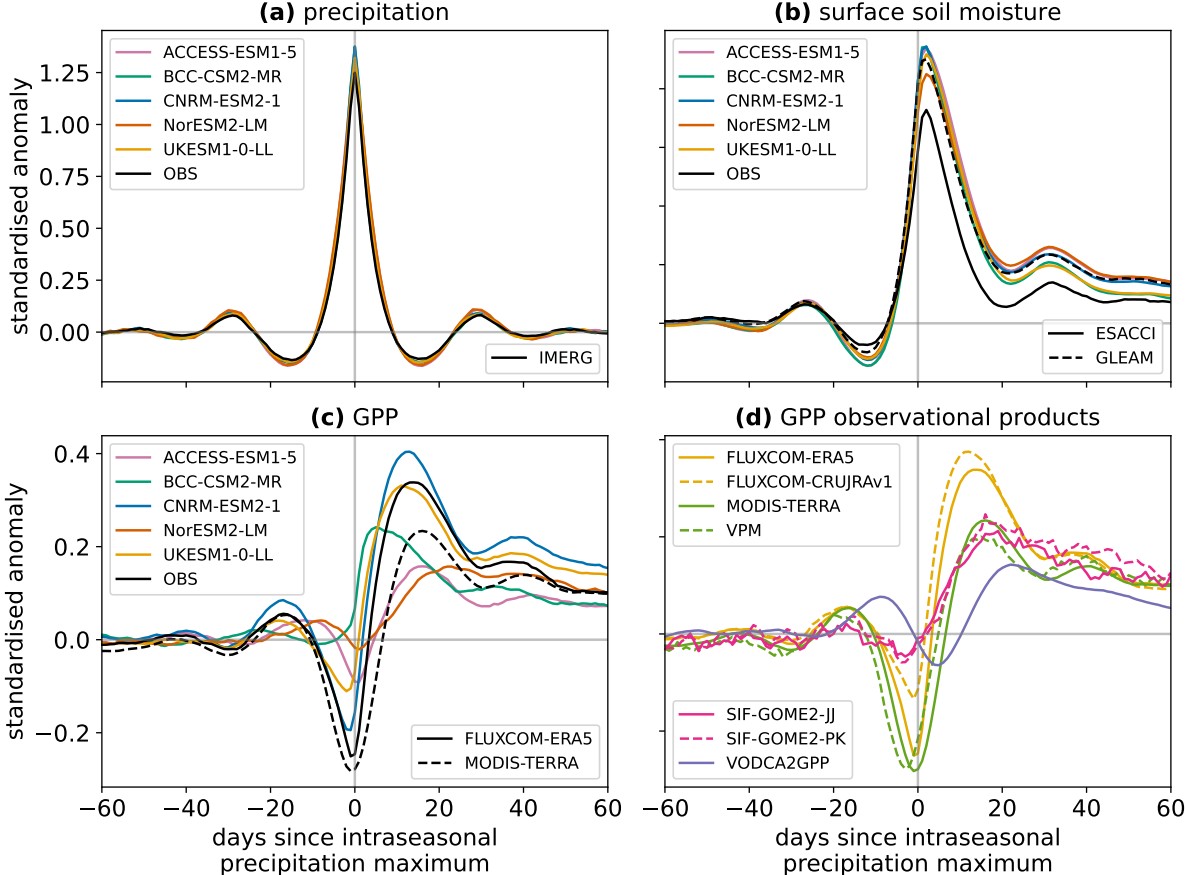

**Figure 1.** Composites of (a) precipitation (b) surface soil moisture and (c, d) GPP around intraseasonal wet events. All land pixels between 60°S and 80°N are included, except in months where the ground is frozen or in regions without valid ESA CCI soil moisture observations. The surface soil moisture and GPP composites have been scaled by the maximum amplitude of their corresponding precipitation composite.

between the types of observational products, with anomaly magnitudes and timings covering most of the behaviour seen in the models. The exception to this is the faster response of BCC-CSM, for which we do not find any observational evidence.

It is therefore clear that there is great uncertainty in the response of GPP to intraseasonal rainfall events in CMIP6 Earth System Models, but that the observations also fail to constrain to this response. This makes it challenging to assess which of the model responses are most realistic from a global perspective. In order to further investigate whether the models are providing appropriate GPP responses, we therefore move to a regional evaluation.

## 3.2 Regional evaluation

To provide a regional assessment of GPP responses to intraseasonal rainfall events, we create composites for each IPCC AR6 land region (Iturbide et al., 2020) using the same method as in section 3.1. This gives 46 regions, which are designed to





represent areas with consistent climate features (Iturbide et al., 2020). Only 42 regions are included in this analysis: the two Antarctic regions have no modelled GPP values, and we remove the desert regions SAH (Sahara) and ARP (Arabian Peninsula) because these show outlying relationships between modelled and observed GPP anomalies, which is likely to be due to the very low GPP in these regions. In order to summarise the GPP responses in each region, we focus on three key properties of the composites, which are illustrated in Figure 2a. The *peak amplitude* in GPP is defined as the maximum standardised anomaly

occurring after the peak of the rainfall event composite (i.e. after day 0). The *lag* is the number of days after the rainfall peak that this peak amplitude occurs. The *post-event amplitude* is the mean standardised anomaly over days 40–60 after the rainfall peak. The amplitude metrics characterise the contribution of intraseasonal wet events to GPP variability, while the lag measures how quickly GPP responds to a wet event.

Figure 2 compares the modelled peak amplitude, lag, and post-event amplitude of GPP in each AR6 region to values from one

of the observation-based products. The observational product used for this comparison is FLUXCOM RS+METEO driven by CRU JRA v1 reanalysis (henceforth referred to as FLUXCOM-CRUJRAv1). This product is chosen as a starting point because FLUXCOM RS+METEO provides the only available daily GPP data, but other observational products will be analysed later in this section. The peak and post-event amplitudes of GPP are scaled by the peak and post-event amplitudes in surface soil moisture respectively (i.e. the modelled peak amplitude in GPP is multiplied by the ratio between the observed and modelled

peak amplitudes in surface soil moisture), since the GPP amplitude is strongly linearly related to the surface soil moisture amplitude (see analysis later in section). ESA CCI soil moisture is used as the benchmark observation. This accounts for the possibility that individual models may under- or overestimate the regional surface soil moisture perturbation at the regional scale following wet events.

All the analysed CMIP6 models show positive correlation for the peak and post-event amplitudes of regional GPP responses

with FLUXCOM-CRUJRAv1, significant at the 95% level (Figures 2b and 2c respectively). Importantly, this indicates that the models are generally able to correctly represent which regions develop larger anomalies in GPP following intraseasonal rainfall events. However, it is clear that the actual modelled values of response amplitude are very different between models. For example, in regions that show higher peak amplitudes in FLUXCOM-CRUJRAv1, UKESM is able to represent these higher amplitudes, whereas the amplitudes modelled by ACCESS-ESM and NorESM are much lower.

The lag of modelled regional GPP responses (Figure 2d) is significantly correlated with observed lags for ACCESS-ESM, BCC-CSM, CNRM-ESM and UKESM. No significant linear relationship is found between observed and modelled lags for NorESM, indicating that the timing of GPP responses to rainfall may not be realistic in this model. In many regions, the GPP lag modelled by NorESM is much longer (20–60 days) than the observed lag in any region. This suggests that the NorESM GPP response is generally too slow, which is consistent with this model showing the slowest response in the global composites

in Figure 1. Even for the models where a significant correlation in lag is found, the correlation coefficients are much lower than for the amplitude relationships, showing that the models match the observations better for the magnitude of the GPP response than the timing.

Given the uncertainty in observed GPP responses established in section 3.1, it is important to assess the effect of changing the observational GPP product on the results of the assessment carried out in Figure 2. The Taylor diagrams presented in





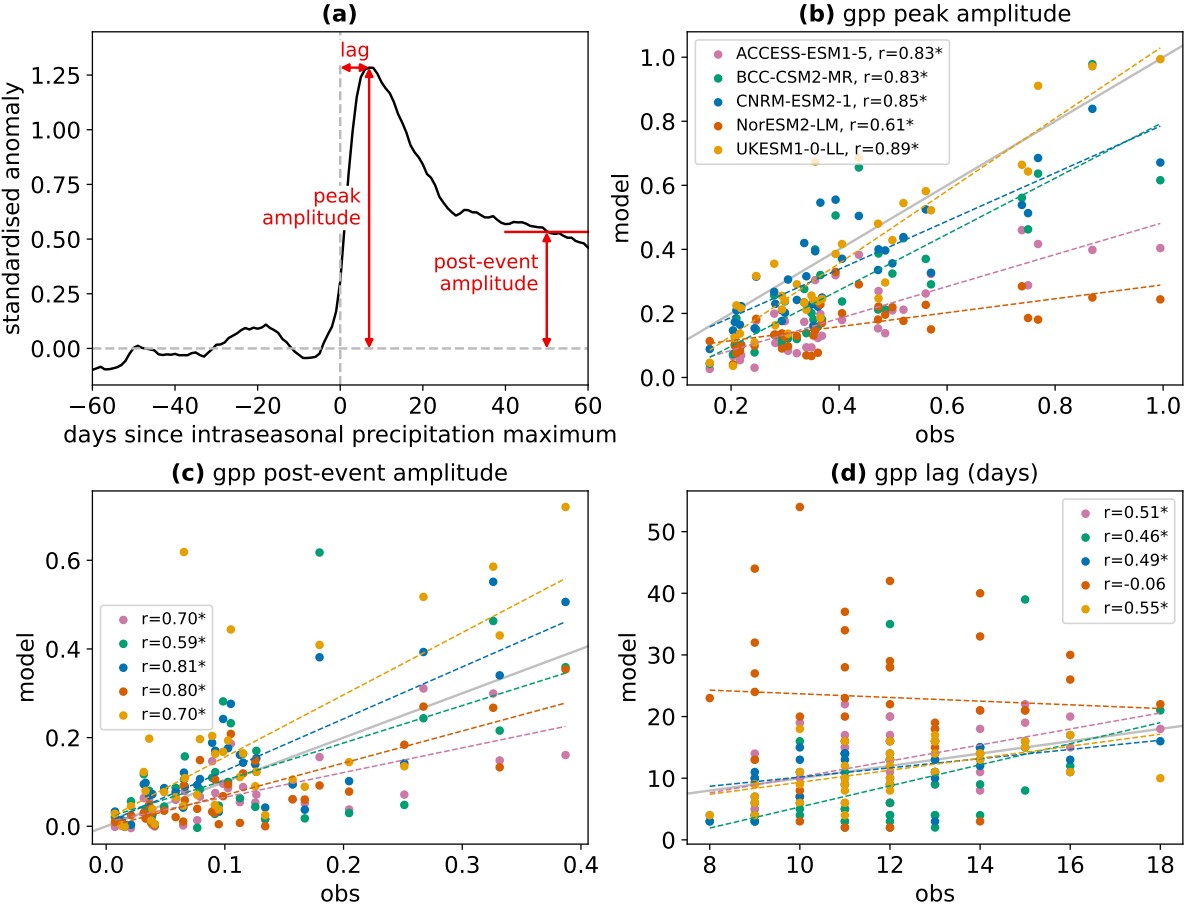

**Figure 2.** (a) Illustration of the three properties of the GPP response composite that will be analysed. The example composite used for the illustration is the GPP response from UKESM in the northern Australia region. (b, c, d) Comparison of the peak amplitude, post-event amplitude, and lag of the GPP response to intraseasonal wet events between the CMIP6 models and the FLUXCOM-CRUJRAv1 observational product. Each scatter point represents an AR6 region. Dashed lines show linear best fits and panel legends detail the linear correlation coefficients. Asterisks after correlation coefficients denote significance at the 95% level. The grey lines indicate the identity line in each panel, i.e. where agreement between modelled and observed values would occur. The peak and post-event amplitudes in GPP have been scaled by the peak and post-event amplitudes in surface soil moisture respectively.

Figure 3 show the correlation between regional responses for each possible model-observation pair, in addition to the ratio of the inter-regional standard deviations ($\frac{\sigma_{\mathrm{model}}}{\sigma_{\mathrm{obs}}}$) and the root mean squared error normalised by the observed standard deviation. For example, in Figure 3a, the orange cross summarises the linear relationship between UKESM and FLUXCOM-CRUJRAv1 that is shown in full in Figure 2b, which has a correlation of 0.89 and a slightly higher spread in peak GPP amplitudes between regions for the model than the observations (i.e. $\frac{\sigma_{\mathrm{model}}}{\sigma_{\mathrm{obs}}} \approx 1.2$).





When focusing on the peak amplitude of the GPP response, the correlation between the model and observations shows some variation based on the choice of observational product, but the value typically lies between 0.8 and 0.9. The exception is NorESM, which is consistently a worse match for the observed regional distribution of GPP response amplitudes (r≈0.6), regardless of the observational product selected. The extent to which the inter-regional standard deviation in peak amplitudes matches the observed standard deviation is much more dependent on the choice of model than on the choice of observational

product. This reflects that some models produce much higher maximum regional peak amplitudes than others (as seen in Figure 2). NorESM produces the lowest inter-regional variation in GPP response amplitudes, related to its inability to represent the larger responses; $\sigma_{\mathrm{model}}$ is far too low compared to the observed standard deviation, regardless of product. ACCESS-ESM also underestimates the inter-regional variation in response amplitudes compared to the observational products, while UKESM overestimates it. CNRM-ESM and BCC-CSM both reasonably capture the variation, falling within the spread of

the observational products. Combined with their high correlation coefficients ($r > 0.8$), these two models therefore provide the most realistic representation of the differences between regions in the peak amplitude of GPP responses to intraseasonal rainfall events.

    The correlation between models and observations is more dependent on the choice of observational product for the post-event amplitude (Figure 3b — note the change in the radial axis scale from Figure 3a) than for the peak amplitude. This

indicates inter-product disagreement on which regions experience elevated GPP in the months following wet events. CNRM-ESM outperforms the other models here in terms of correlation, and also produces a standard deviation consistent with several of the observational datasets. Similarly to the results for the peak amplitude, NorESM and ACCESS-ESM tend to underestimate the inter-regional variation in response, while UKESM overestimates it. In other words, intraseasonal rainfall events in NorESM and ACCESS-ESM play too minor a role in GPP variability compared to in observations.

Whereas all pairs of models and observations show correlations with $r > 0.5$ for both the peak and post-event amplitudes, the modelled lags in the regional GPP responses show much less agreement with the observed lags (Figure 3). Many of the relationships do not show a correlation significant at the 95% level, with some even having a negative correlation. For each model, there are also large spreads in both the correlation and $\frac{\sigma_{\mathrm{model}}}{\sigma_{\mathrm{obs}}}$ dependent on the observations used for the assessment. The timing of GPP responses to intraseasonal rainfall events is much less well constrained by observations than the amplitude.

We now consider what may be causing such large differences between models. One possible reason for differences between models in their GPP responses to water availability is the presence of discrepancies in their land cover maps. However, inter-model differences in the amplitude and lag of GPP responses can be seen even in regions where the models have similar land cover (see Appendix A). Therefore, the differences result from variations in the models' representation of processes linking water availability and vegetation productivity. Two likely candidate processes with known uncertainty in ESMs are the

control of soil moisture stress on GPP and the response of stomatal conductance to changes in VPD. We therefore investigate the regional relationships between the perturbations in surface soil moisture (SSM) and near-surface vapour pressure deficit (VPD) and the GPP response following the wet events. Figure 4 explicitly compares these relationships in each model and in observations. Again, FLUXCOM-CRUJRAv1 is used as an illustrative observational GPP product, but the other products will be included later in the section to test the robustness of the relationships. ESA CCI Soil Moisture is used for the SSM





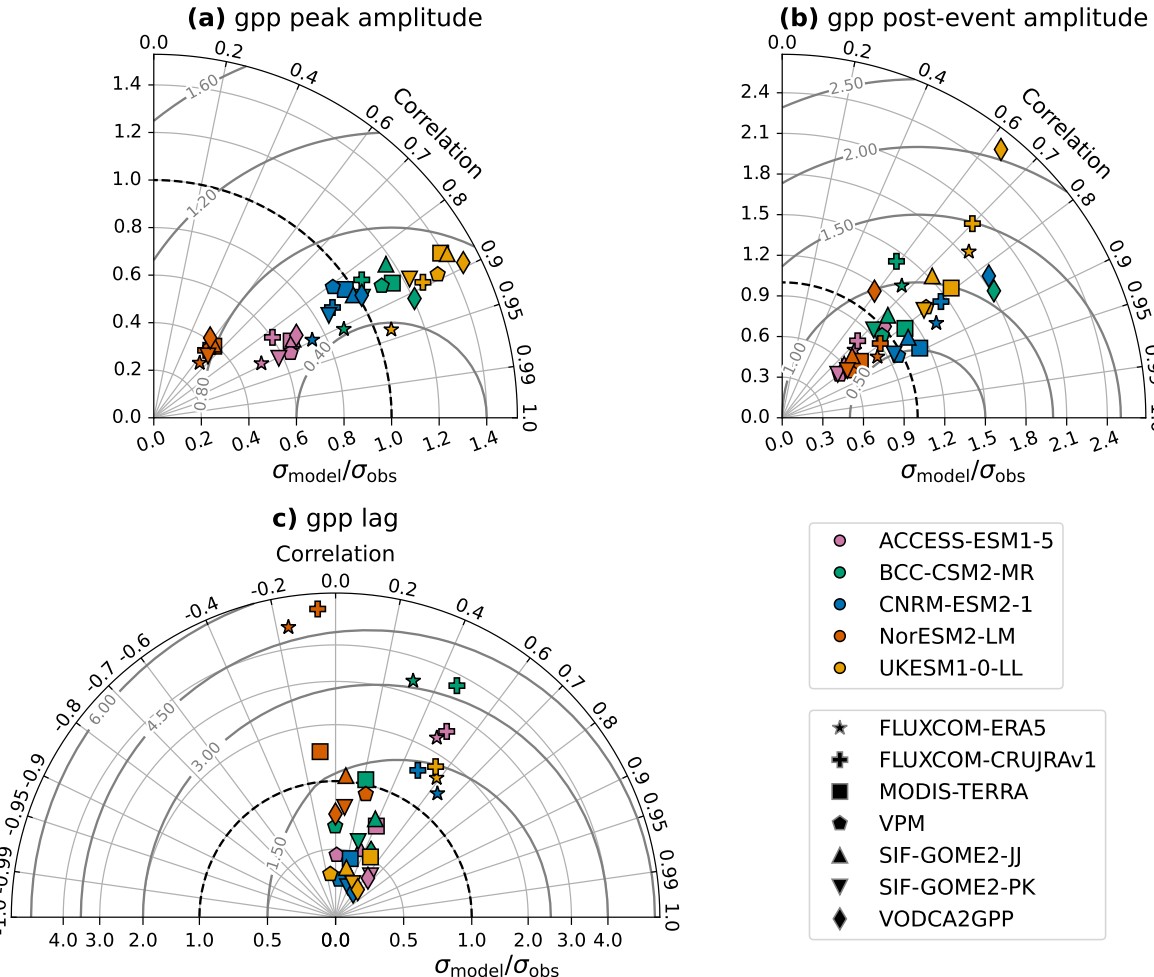

**Figure 3.** Taylor diagrams of regional relationships between modelled and observed GPP responses. Markers show the correlation and ratio of standard deviations $\frac{\sigma_{model}}{\sigma_{obs}}$ between the model and each observation-based GPP product for the (a) peak amplitude, (b) post-event amplitude, and (c) lag of the GPP response to intraseasonal wet events. The colour of the marker indicates which model is being evaluated, and the shape of the marker indicates the observation-based product used. The labelled grey contours indicate the root mean squared error normalised by observed standard deviation. GPP peak and post-event amplitudes have been scaled by the modelled peak and post-event amplitudes in surface soil moisture respectively.

observations, but the results are qualitatively unchanged if it is substituted with GLEAM (not shown) . When defining the peak amplitude of the VPD response, the minimum rather than the maximum standardised anomaly is taken, since VPD typically decreases during and after intraseasonal wet events. Note that daily VPD data was not available for BCC-CSM, so this model is not included in the VPD analysis. SSM and VPD perturbations are not independent from one another; the regional amplitudes



of SSM and VPD responses are strongly negatively correlated. However, we test the relationship of GPP with both variables
since they influence the GPP through different modelled processes.

All models represent the observed positive correlation between the SSM peak amplitude and the GPP peak amplitude: re-
gions with a larger standardised anomaly in SSM following rainfall events show a larger standardised anomaly in GPP, due to
the greater change in water availability. The models also capture the observed negative correlation between the peak amplitudes
of the VPD and GPP responses. Regions with a smaller drop in VPD standardised anomaly after wet events are experiencing
less evaporative demand compared to climatological conditions, so stomata can remain open to allow photosynthesis with-
out experiencing as much water loss to the atmosphere. These correlations are significant ($p < 0.05$) in all the observational
products tested. However, even though all models represent these basic relationships, there are large differences between the
correlation coefficients, meaning that GPP responses to rainfall are more tightly coupled to the associated perturbations in
SSM and VPD in some models than others. For example, the peak amplitude of the SSM perturbations can explain 90% of the
variance in the GPP peak amplitude between regions in CNRM-ESM2-1, whereas in UKESM1-0-LL it only explains 57%.

There are also differences between models in the slopes of the linear relationships. This demonstrates that the models' GPP
peak amplitudes have varying sensitivities to changes in the peak amplitude of the SSM/VPD perturbations. For example,
for a given increase in the peak standardised anomaly of SSM, NorESM exhibits a much smaller increase in the associated
GPP peak anomaly than the other models. We test the significance of these differences in slope among all the models and
all the observational GPP products. This is done using estimated marginal means of linear trends and correcting for multiple
testing using the Tukey method. This analysis shows that for the relationship between the peak amplitudes of SSM and GPP,
the sensitivity of NorESM is significantly different ($p < 0.05$) to the sensitivity of all other models and all observational GPP
products. There are no other pairs with a significant difference in slope. Therefore, we conclude that the amplitude of GPP
responses to wet events in NorESM is not sufficiently sensitive to the changes in surface soil moisture.
The peak amplitude of GPP responses is also less sensitive to VPD perturbations in NorESM than in the other models, but in
this case the slope of the relationship in NorESM is significantly different only to UKESM and FLUXCOM-ERA5. UKESM
also shows significantly different VPD sensitivity to ACCESS-ESM and to MODIS Terra. There are clearly differences between
models, but there is insufficient evidence here to state that any are unrealistic compared to the range of observations.

Similar relationships between SSM/VPD and GPP are also found for the post-event amplitudes, but the correlation is gener-
ally weaker than for the peak amplitudes. The peak amplitude of GPP is more strongly constrained by the concurrent SSM and
VPD anomalies than the post-event amplitude. In the case of NorESM, the correlation is sufficiently weak that no significant
relationship is found between SSM and GPP amplitudes after the event, when testing at the 95% level; vegetation productiv-
ity is much less strongly linked to surface soil water conditions than in the other models and in the observations (again, the
correlations are significant ($p < 0.05$) in all the observational products tested). The weaker relationships between post-event
amplitudes can be partly explained by the fact that over the 40–60 day timescale, changes in root zone soil moisture explain
more of the variation in the GPP amplitudes than changes only in the surface layer (whereas the surface layer is generally
more important for the peak amplitude). Comparing the post-event amplitude of soil moisture in the top 1 m of soil with the
post-event amplitude of GPP gives higher correlation coefficients than seen in Figure 4c for all models (other than BCC-CSM,





for which the 1m-layer data is not available), as shown in Figure S3. In the months after the peak of the wet event, the rainfall
infiltrates the soil and provides additional moisture availability to vegetation at deeper soil levels. Although Figure 4c shows
that ACCESS-ESM and NorESM are less sensitive to the SSM post-event amplitude, there are no statistically significant inter-
model differences in slope, and only a few significant differences in slope between the models and observations. The modelled
sensitivity to VPD post-event amplitude is not inconsistent with observations for any model.

The lag in GPP response is compared to the peak amplitude rather than the lag of SSM/VPD, because there is very little
variation in SSM/VPD lags between regions due to these variables responding very quickly (1–3 days) after rainfall. In the
models, the lag of the GPP response tends to be shorter in regions with larger perturbations in SSM and VPD. All models
exhibit this relationship except NorESM. However, the evidence for the relationship in the observations is much weaker (see
Figure S2). Only two of the seven observational products tested show a significant negative correlation between SSM peak
amplitude and GPP lag (FLUXCOM-ERA5 and VODCA2GPP), while VPM shows a significant positive correlation. Four out
of the seven products show a significant positive correlation between VPD peak amplitude and GPP lag (FLUXCOM-ERA5,
FLUXCOM-CRUJRAv1, SIF-GOME2-PK and VODCA2GPP). Even in the observational products that do show these signif-
icant relationships, the correlation coefficients are much lower than those obtained from the models. The highest correlation
coefficients from observations are from VODCA2GPP, at $r = -0.51$ for the correlation between SSM peak amplitude and GPP
lag and $r = 0.56$ for VPD peak amplitude and GPP lag. These correlations are considerably weaker than those found in the
models (other than NorESM), which have $|r|$ between 0.6 and 0.8. It is therefore possible that the timing of the GPP responses
in the models is too tightly constrained by the amplitude of the regional perturbations in SSM and VPD.

## 4 Discussion and Conclusions

This work has demonstrated that the responses of vegetation productivity to intraseasonal rainfall events are very different
between ESMs participating in CMIP6. Of the models tested, UKESM and CNRM-ESM tend to produce larger standardised
anomalies of GPP following rainfall events, while ACCESS-ESM and NorESM produce smaller productivity anomalies. There
are also differences in the timing of the models' responses following the rainfall events, with NorESM taking much longer to
reach its peak GPP anomaly, and BCC-CSM responding most quickly. The models all correctly represent that larger anomalies
in regional GPP are associated with larger increases in surface soil moisture and larger decreases in near-surface vapour pressure
deficit. However, the strength of these relationships and the sensitivity of the GPP responses to these drivers varies between
models. In particular, GPP in NorESM is not sufficiently sensitive to intraseasonal surface soil moisture perturbations. This
aligns with the findings of Anav et al. (2015) that NorESM has lower seasonal and interannual variability than other models.

The reasons for this diversity in model behaviour are likely multiple and complex. We suggest that unravelling them requires
an in-depth knowledge of each model configuration, parameterisation structure and parameter values, which is beyond the
scope of this paper. Likely causes include differences in the soil moisture stress function applied to down-regulate GPP, and
in the stomatal conductance models, which control how changes in VPD affect GPP, as discussed in section 1. Further work
should therefore compare the GPP responses between different versions of particular ESMs or land surface models to establish



**Figure 4.** Regional GPP responses to intraseasonal wet events compared to the driving perturbations in surface soil moisture (a, c, e) and vapour pressure deficit (b, d, f). Each scatter point represents an IPCC AR6 region. Dashed lines show linear best fits. The legend for each panel indicates the linear correlation coefficient between the driving perturbation and the GPP response for each model. Asterisks after correlation coefficients denote significance at the 95% level. The uppermost legend, showing which model is denoted by each colour, applies to all panels. The observational products used are FLUXCOM-CRUJRAv1 (GPP), ESA CCI (surface soil moisture) and ERA5 (VPD).





how specific changes in configuration—for example, alternative soil moisture stress parameters and stomatal conductance models—impact the coupling between water availability and vegetation productivity. The vertical profile of root water uptake may also play a role. Each model discretises the soil column into layers differently and prescribes different root depths for plant functional types. In a model where vegetation is able to access deeper reserves of soil water, it may become less water-stressed and therefore exhibit smaller responses to rainfall events. Additionally, the partitioning of rainfall into direct evaporation, transpiration and runoff, which operate on different timescales, will impact the magnitude and timing of water availability for vegetation following a rainfall event. The post-rainfall GPP response could also be affected by each model's representation of drought deciduous phenology (for example, drought deciduous phenology is turned off in UKESM).

Further work is therefore needed to understand why different methods for deriving global GPP products result in different relationships with water availability, quantify the uncertainty in these products, and ultimately to obtain observations that will reduce our uncertainty in the response of GPP to intraseasonal rainfall events. Some of the issue with timing in the observations may be due to the 8-daily temporal resolution of all products other than FLUXCOM RS+METEO. We therefore emphasise the usefulness of GPP data at the daily timescale for probing process-based diagnostics of climate-carbon cycle coupling. Insights at this timescale are key for constraining model processes, and in many regions the response of vegetation productivity to events on this timescale is important for determining its annual mean (e.g. Wu et al., 2021). Daily SIF observations could be a valuable resource for understanding GPP responses to rainfall events, particularly since they are less affected by cloud cover than alternative vegetation observations such as NDVI.

This study has demonstrated a framework for evaluating an important link between the water and carbon cycles in ESMs by assessing the response of GPP to intraseasonal rainfall events. In the recent historical period of CMIP6 simulations, models show diverse behaviour in their GPP responses, and only some aspects of this can be constrained by the currently available observations. It is therefore difficult to determine which models represent the most realistic vegetation responses to wet conditions. This means it is challenging to understand and model the behaviour of the land carbon sink and its consequences for atmospheric $CO_2$ levels in the present day, and to have confidence that the sink is represented correctly in projections under future scenarios. Improvements are required both in our understanding of how model configurations and parameters affect the resulting vegetation productivity responses to water availability, and in the observational datasets used to evaluate this coupling.

*Code and data availability.* Daily GPP data from FLUXCOM RS+METEO is available on request from the FLUXCOM team (see http://www.fluxcom.org/CF-Download/). All other datasets used in this paper are publicly available for download. CMIP6 model data: https://esgf-node.llnl.gov/projects/cmip6/ (accessed 8th February 2023). ERA5: https://cds.climate.copernicus.eu/cdsapp#!/home (accessed 1st March 2023). IMERG precipitation: https://doi.org/10.5067/GPM/IMERGDF/DAY/06 (Huffman et al., 2019). ESA CCI soil moisture: https://www.esa-soilmoisture-cci.org combined product v06.1 (accessed 23 July 2021). GLEAM v3.6a soil moisture: https://www.gleam.eu (accessed 16 December 2022). MODIS Terra GPP aggregated to 0.5° resolution: https://doi.org/10.25592/uhhfdm.8556 (Kern, 2021). VPM GPP (8-daily at 0.5° resolution): https://doi.org/10.6084/m9.figshare.c.3789814.v1 (Zhang et al., 2017b). Downscaled GOME-2 SIF: http://data.europa.eu/89h/21935ffc-b797-4bee-94da-8fec85b3f9e1 (Duveiller et al., 2019). VODCA2GPP: https://doi.org/10.48436/





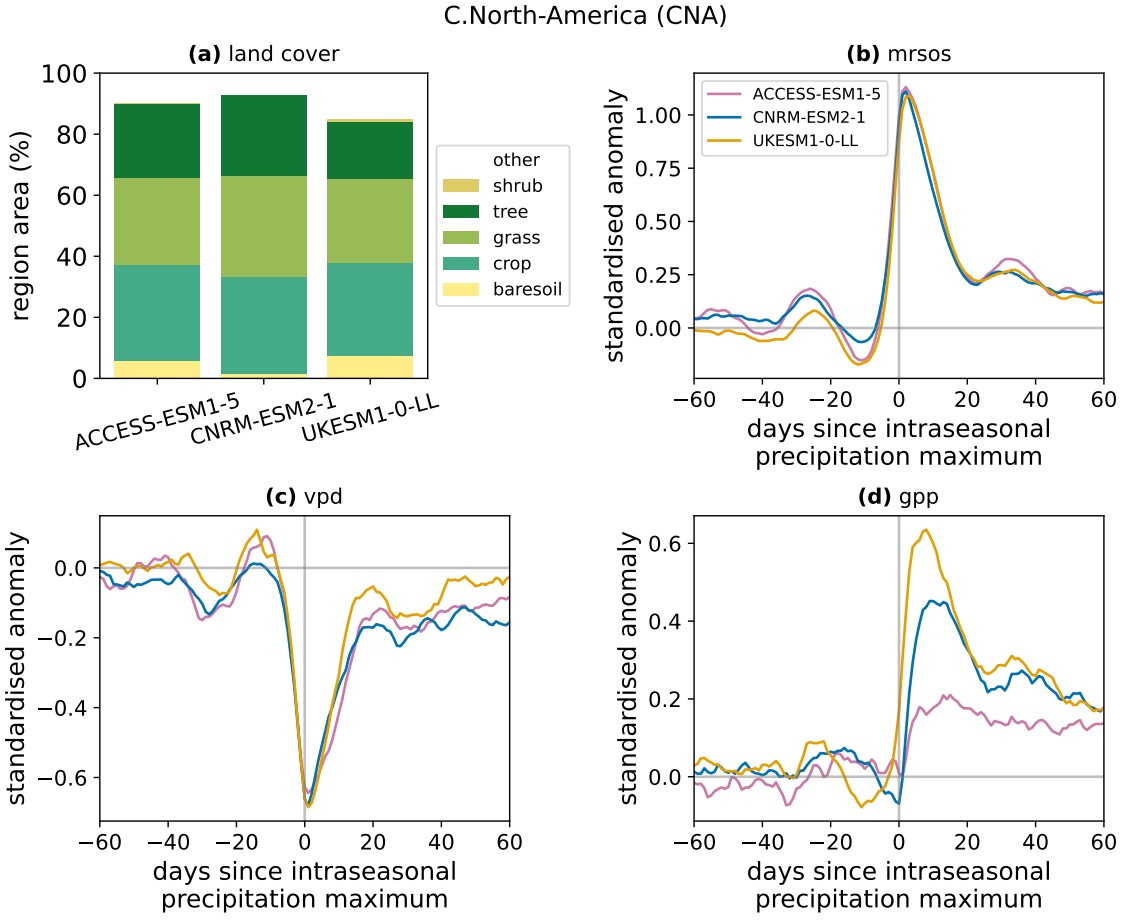

**Figure A1.** Land cover and responses to intraseasonal wet events in the Central North America (CNA) AR6 region. (a) Percentage of the region assigned to each land cover. The "other" category includes water bodies. (b, c, d) Responses of standardised anomalies in surface soil moisture (b), near-surface vapour pressure deficit (c) and GPP (d) in the three models that provide land cover information. These composites have been scaled by the maximum of the precipitation composite for each model.

1k7aj-bdz35 (Wild et al., 2021). The code used to generate the results and figures in this paper is available at https://github.com/bethanharris/cmip6-gpp-isv.

**Appendix A**

Inter-model differences in the GPP responses to water availability could arise due to the presence of discrepancies in the models' land cover maps. For example, if one model represents a higher percentage of forested land in a region, its GPP is

likely to show less of a response to intraseasonal rainfall events in this region than a model with higher grassland coverage.

To demonstrate that differences in GPP responses between models occur even where the land cover distributions are similar, Figure A1 shows the land cover distributions and the responses of SSM, VPD and GPP to intraseasonal rainfall events (as in the global composites of Figure 1) for the IPCC AR6 Central North America (CNA) region (Iturbide et al., 2020). The mean land cover over the period 2000–2014 is used for Figure A1a. Only three models are included: land cover data was not available for
BCC-CSM2-MR or NorESM2-LM.

All three models have similar land covers in this region. ACCESS-ESM and UKESM are the most alike, with CNRM-ESM having a lower fraction of bare soil and more vegetated cover. However, this higher similarity between the land covers of ACCESS-ESM and UKESM does not translate into greater agreement in their responses of GPP to rainfall. Although the models produce consistent responses in SSM and VPD, the GPP responses are very different, as was found in the global
evaluation of section 3.1. The peak standardised anomaly in GPP following rainfall events in this region is approximately three times higher in UKESM than in ACCESS-ESM. Since in this case we have established that the difference is not due to a disparity in the land cover distribution, we conclude that it is the result of differences in the models' representations of water-vegetation coupling processes.

*Author contributions.* BLH: Conceptualization, data curation, formal analysis, investigation, methodology, software, visualization, writing
– original draft preparation, writing – review & editing. TQ: Funding acquisition, writing – original draft preparation, writing – review & editing. CMT: Conceptualization, funding acquisition, supervision, project administration, writing – review & editing. PPH: Methodology, writing – review & editing.

*Competing interests.* The authors declare that they have no conflict of interest.

*Acknowledgements.* This study was funded as part of the Natural Environment Research Council's support of the National Centre for
Earth Observation, via the projects CPEO (Constraining Coupled Carbon & Water Cycle Processes with Earth Observation), Grant No. NE/X006328/1, and TerraFIRMA (Future Impacts, Risks and Mitigation Actions in a changing Earth system), Grant No. NE/W004895/1. We acknowledge the World Climate Research Programme, which, through its Working Group on Coupled Modelling, coordinated and promoted CMIP6. We thank the climate modeling groups for producing and making available their model output, the Earth System Grid Federation (ESGF) for archiving the data and providing access, and the multiple funding agencies who support CMIP6 and ESGF. We thank
the FLUXCOM team for making the daily GPP data from FLUXCOM RS+METEO available to us. The figures were designed using the colourblind-safe colour scheme provided by Wong (2011). The Taylor diagrams in Figure 3 were produced using an adaptation of code from Copin (2012).



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
