# Peer review of "Contrasting responses of vegetation productivity to intraseasonal rainfall in Earth System Models"

_Earth System Dynamics, 2024_

## Author Comment (AC1)

**Contrasting responses of vegetation productivity to intraseasonal rainfall in Earth System Models**

**Response to reviewers: 2024/04/23**

We would like to thank both reviewers for their time spent reviewing the manuscript and for raising important points. All reviewer comments are addressed below and our proposed changes to the manuscript detailed. Reviewer comments are in blue italics with our responses following in black. Proposed new text for the revised manuscript is in red.

**Reviewer 1**

*1 General Comments:*

*I recommend a revise and re-submit. This paper contributes significantly to our understanding of the impact of precipitation events on carbon cycles, particularly from the perspective of individual events using Earth System Models (ESMs). However, there are some concerns that the authors need to address before the paper can be published.*

*This paper explored how Earth System Models (ESMs) perform in response of vegetation productivity to individual precipitation events. Papers in remote sensing and model simulations usually focus on the interannual change of precipitation and its impact on vegetation productivity, and they have paid little attention to the properties and impact of individual precipitation events. However, manipulated experiments are carried out by monitoring vegetation growth after every rainfall, providing plenty of data about the individual precipitation events. We have limited knowledge of how ESMs simulate vegetation responses after rainfalls. Investigating this question can improve ESMs by combining findings in manipulated experiments. On the other hand, the research on ESMs could also provide an analytical framework for experiments.*

*So, this paper is a great attempt to research individual precipitation events. They found distinct simulations in different models by composite precipitation events together over regions. They focused on three propensities: Peak amplitude, post amplitude, and time-lag effect. They didn't discuss the amount of GPP impacted but used the sensitivity of GPP to precipitation instead.*

*During their discussion, they explored various reasons for the discrepancies between different models. They narrowed their focus to the processes that related to soil moisture and VPD, which link vegetation productivity and water availability. By comparing model simulations with observations, they found models perform badly in simulating GPP lag and its relationship with soil moisture. Their work told us the timescale of vegetation response should be the main aspect of model improvement.*

*I highly appreciate that they have thoroughly tested their results by using data from multiple sources and reporting the uncertainty caused by the data sources. This also revealed the inconsistency between observations datasets of vegetation productivities.*

Thank you for your positive review of the work. We would like to clarify that the precipitation events around which we composite are not necessarily "individual precipitation events" as such, but wet intraseasonal periods with a characteristic timescale of ~1 month. A wet intraseasonal period may in fact contain multiple individual shorter rainfall events. In some cases it will indeed be a single precipitation event, which is large enough to cause that intraseasonal period to have unusually

elevated rainfall overall. By using this method, rather than focusing on true individual precipitation events, we guarantee that we will obtain a driving precipitation composite with the same timescale for each model/observations (i.e. all composites in Figure 1a are extremely similar to one another, with precipitation anomalies of the same duration). This means we can fairly compare the timescales of the GPP responses between the models and the observations, without the need to consider differing durations of rainfall events. We have included more detail on the method of event identification and how we will clarify the definition of the events in response to a comment later in the review.

*2 Specific Comments:*

*There are some concerns that I think must be handled before re-submit. Firstly, The spatial scale of the precipitation events must be considered. Second, the sample size of the composite should be provided. Addressing these two concerns is essential before resubmitting the manuscript.*

*The critical assumption they made is that GPP dynamics in a 1 x 1 degree grid box can be a representation of vegetation response to individual precipitation events. However, the scale of precipitation events is highly variable, ranging from 10km to 400 km (Matte et al., 2022; Post et al., 2021). Moreover, spatial variation of daily precipitation is high even on a 2-4 km scale (Augustine, 2010), let alone on a 110 km scale (1 x 1 degree scale). So, if the spatial resolution of GPP data does not match the spatial scale of precipitation events, it is hard to contribute the GPP anomaly to the individual events. Additionally, using the observation data that aggregated to a coarse resolution, the precipitation event can occur on any part of the grid box, impacting different land covers and causing varied responses of vegetation productivity. So, a supplementary analysis is necessary to show the robustness of the results in Figure 1. The sensitivity test of spatial resolution for aggregation results in Figure 1d. The results in Figure 1d would be highly variable using different spatial resolutions aggregating observation data. If the effect of spatial scale is insignificant, the variation caused by spatial resolution will be less than that from different GPP observation products.*

*If the supplementary analysis is not possible, some references could be the substitute to demonstrate that 1 x 1 degree grid box has the ability to reflect the vegetation responses to individual precipitation events. For example, Post et al. (2021) found the vegetation response to extreme precipitation can be scaled up to 10 km. However, Zhang et al. (2023) found the coarse-resolution remote sensing images overlooked many greening areas driven by interannual precipitation change.*

Many thanks for raising this really important point. Since our primary aim in this work was to evaluate the performance of CMIP6 Earth System Models, which represent atmospheric and land processes at scales on the order of 100km, it is natural for us to analyse the observations at the same scale. However, we agree that it is necessary to establish that the observed relationships are valid at smaller scales within the coarse grid boxes rather than simply assuming that the precipitation and GPP changes are collocated. We have therefore performed the suggested analysis using the VPM GPP dataset as an example, since this dataset is made available on a 0.05° grid. (The FLUXCOM RS+METEO dataset, for example, is produced at 0.5°, so it would not be possible to gain any insight into its behaviour at a resolution any finer than this.) The IMERG precipitation dataset is produced at 0.1°. We therefore reproduce Figure 1d using data at a resolution of both 0.1° and 0.25° and compare this to the original 1° analysis included in the manuscript. Intraseasonal wet events are defined at the finer resolution using an identical method to the original 1° approach. Figure R1 below shows the results of these analyses.

[Figure]

*Figure R1: Response of VPM GPP to intraseasonal wet events at varying spatial resolutions. The legend in (b) shows the spatial resolution of data used and n, the maximum number of GPP observations being composited on any particular day around the intraseasonal maximum.*

The variation in the GPP response between different resolutions is small—much smaller than the difference in response between the observational products. This indicates that the method we use to explore GPP responses in this work is not sensitive to spatial resolution and therefore it is suitable to use the observational responses to capture the average behaviour of a coarse grid box, as the Earth System Models are designed to do.

We will include Figure R1 as an additional supplementary figure in the revised manuscript, and add the following text to Section 3.1:

*Whilst a horizontal resolution of 1° is relevant for evaluating ESMs, which represent an average coarse-scale behaviour of the earth system, it is possible that the intraseasonal precipitation events and the vegetation responses do not spatially co-occur within this 1° grid box in observations. Therefore, to test whether the 1° GPP responses are attributable to the precipitation anomalies, we also composite the GPP response around intraseasonal wet events using the IMERG and VPM datasets at 0.25° and 0.1° horizontal resolution. The global composite GPP response is not found to be sensitive to spatial resolution (Figure S1). This indicates that the coarser 1° response, which represents the vegetation response at a scale relevant to ESMs, is representative of the processes occurring at much smaller spatial scales in the real world.*

We also compared the GPP response at 0.1° to the 1° response for each IPCC AR6 region, to check that this assertion held for each region individually. Figure R2 shows that the response shows low sensitivity to spatial resolution for all regions. We do not intend to include Figure R2 with the final manuscript but include it here for your interest.

We note that the insensitivity to spatial scale is likely to be related to the method we are using to define the intraseasonal wet periods (more details later in response): since an event must show a precipitation anomaly at least 1σ above the (intraseasonally filtered) mean at the 1° scale, it is unlikely that a very localised single storm would achieve this threshold, whereas widespread anomalous rainfall across the grid box over multiple days is more likely to do so.

[Figure]

*Figure R2: GPP responses to intraseasonal precipitation events at varying spatial resolution, as in Figure R1 but decomposed into each IPCC AR6 region.*

*Secondly, I am wondering about the sample size used to create the composite over regions. The data was from 2000-2014 and there might be many precipitation events in one year. However, due to the coarse spatial resolution, the number of grid boxes in one region might not be that big. Without the sample size provided, it is difficult to have confidence in the results. It would be helpful if you could provide some sample sizes in the results or method section, where the composite is introduced.*

We have created a new figure to show the sample size per region (Figure R3). We will include this as an additional figure in the Supplementary Information and note at line 203 (when the regional composites are introduced) that:

All regions included in the analysis have a minimum of 500 precipitation events identified in each model and in observations. Full details of the sample sizes of events are included in Figure S2.

[Figure]

*Figure R3: Sample size of intraseasonal precipitation events in each IPCC AR6 region (Iturbide et al., 2020).*
*(a) Mean number of events in each region across all 5 models and observations. Regions that are excluded from the analysis entirely are marked in grey. Note that in regions crossing the 60 °N latitude line, only events south of 60 °N are included. (b) Comparison of event sample sizes between models and observations for each region. The regions are in ascending order according to the mean number of events as shown in (a). Note the change of y-axis scale between the two panels.*

During the creation of Figure R3, we realised that there was a discrepancy in several high-northern-latitude regions between the number of events produced by the models and the number of events identified in the observations. This is due to poorer data coverage north of 60°N in the IMERG dataset, which occurs because of the lack of infrared observations north of this latitude and icy/snowy surfaces affecting microwave retrievals. These data gaps mean that there are very few intraseasonal precipitation events identified from the observations above 60°N, while a large number are identified from the model data. We will therefore restrict all analyses to the latitude band 60°S–60°N (as opposed to 60°S–80°N in the original manuscript), to ensure that the models are only compared to areas in which precipitation is well-observed. This results in only extremely minor changes to the results presented in the manuscript (some r values in the figures/text will be slightly changed but the identification of significant/non-significant correlations remains the same).

We will change the (60°S–80°N) definition on line 172 to (60°S–60°N), and add an explanation of this choice at line 165:

*The dates of these events are identified separately for each CMIP6 model and for IMERG. Due to gaps in the IMERG dataset at high latitudes, this procedure results in far fewer observed events than modelled events outside the latitude band 60°S–60°N. We therefore restrict all analyses of both modelled and observational data to 60°S–60°N in order to ensure a fair comparison between the two.*

We will also change the text at lines 195–205 as follows:

*To provide a regional assessment of GPP responses to intraseasonal rainfall events, we create composites for each IPCC AR6 land region (Iturbide et al., 2020) using the same method as in section 3.1. This gives 46 regions, which are designed to represent areas with consistent climate features (Iturbide et al., 2020). Only 40 regions are included in this analysis: GIC (Greenland/Iceland), RAR (Russian Arctic) and the two Antarctic regions  lie entirely outside the 60°S–60°N latitude band in which we identify intraseasonal precipitation events. We also remove the desert regions SAH (Sahara) and ARP (Arabian Peninsula) because these show outlying relationships between modelled and observed GPP anomalies, which is likely to be due to the very low GPP in these regions.*

*Finally, I have a suggestion regarding the presentation of the impact on vegetation productivity. Given that the models used the same precipitation scenario, the sensitivity and the total amount of changes in GPP are essentially equivalent. Therefore, I propose that the authors provide the amount of vegetation productivity change caused by individual events. By comparing the total GPP anomaly of composites from different models, we can directly assess which model overestimates or underestimates the carbon assimilation after rainfalls. It gives a clear indicator of whether the model accurately simulates vegetation productivity and allows for comparison with other remote sensing studies in the future. If authors consider sensitivity to be their primary focus, this suggestion can be dismissed.*

We do agree that the total GPP anomaly is an important aspect of rainfall responses. However, we consider our focus here to be the sensitivity and so have chosen not to pursue this.

*Below are some specific questions I want to address, LXXX refer to line XXX in the preprint.*

*L154: Why does this method have the capability to find the date? Maybe it would be better to have a brief introduction to Lanczos filtering.*

We will include more detail on the function of the Lanczos filtering and further explanation of the event identification method (as discussed previously in this response). New text at L155 will be added as follows:

*For each 1°×1° grid box, we apply a 25-day low-pass Lanczos filter to the daily precipitation anomaly, where the anomaly is computed relative to the climatology for a 7-day rolling window.* *This removes variability at a frequency higher than 25 days from the anomaly time series. Intrasesasonal precipitation events are then defined as local maxima of the filtered time series that lie above one standard deviation from the mean.* *This method identifies wetter-than-usual intraseasonal periods with a characteristic timescale of approximately 25 days. The elevated precipitation associated with an intraseasonal wet event may be a single large rainfall event or it may comprise several shorter bursts of rainfall. Using such a filtering approach ensures that the driving precipitation composites have a consistent timescale, so that they provide a fair basis on which to compare the timescales of the GPP responses between the models and the observations.*

To ensure that the timescale of the events is clear, we will also further emphasise this in the abstract at line 2:

*We evaluate this response at the intraseasonal timescale in five CMIP6 ESMs by analysing changes in Gross Primary Productivity (GPP) after intraseasonal rainfall events* *with a timescale of approximately 25 days.  We compare these responses to  those found in a range of observation-based products.*

*L206: The post-event amplitude seems to be too subjective when compared with the other two metrics. I'm curious as to why the researchers chose the time range of 40-60 days for this metric. Was this range based on any references? If they base this range on the paradigm in Figure 2(a), it might not be suitable for regions or models with lag over 30 days (Figure 2d, see the modeled GPP lag of NorESM2-LM). In such cases, the GPP only begins to decline from its peak within the 40-60 day range. Consequently, the time range for the post-event period should be adjusted to account for GPP lags.*

This time range was chosen based on the timescale of the precipitation variations rather than the timescale of GPP behaviour. There is no longer any precipitation anomaly by this time (or only a very small anomaly—as seen in Figure 1a). By "post-event" we simply intended to convey that there is no longer a disturbance in rainfall rather than that vegetation is no longer affected. We will modify the text at line 206 to read:

*The post-event amplitude is the mean standardised anomaly over days 40–60 after the rainfall peak,* *a period when there is no longer an anomaly in precipitation.*

*L213: Why did the GPP be scaled by the ratio of soil moisture? Was it done to eliminate the impact of simulated soil moisture on GPP? By using the scaled GPP, the objective was to measure the sensitivity of GPP to soil moisture instead of precipitation. Readers might be confused. It would be helpful if you could provide an explanation of why the scaling is necessary.*

We agree that this was not explained clearly enough. The scaling was done to allow comparison of the sensitivity of GPP to soil moisture across the different models and observations. The

precipitation composites for a given region do not all have identical amplitudes across the models, due to slight differences in precipitation variability. The amplitudes of soil moisture composites also do not match perfectly across the models (and observations). If the scaling is not applied, then one model may have a higher GPP response amplitude than another in a region because it is being driven by a larger soil moisture perturbation, rather than because the model's GPP response is more sensitive to soil moisture: we consider the latter to be the focus of our research. Essentially, it wouldn't be surprising that models with a higher soil moisture anomaly then give a higher GPP anomaly, but we wanted to show that if you scale them to have the same soil moisture anomaly, the GPP response is *still* different. We chose to scale by soil moisture rather than precipitation because we found a stronger linear relationship between soil moisture and GPP than between precipitation and GPP and therefore soil moisture was more appropriate for a linear scaling.

We will clarify this by modifying the text starting at line 213:

*The peak and post-event amplitudes of GPP are scaled*  *according to the observed peak and post-event amplitudes in surface soil moisture respectively (i.e. the modelled peak amplitude in GPP is multiplied by the ratio between the observed and modelled peak amplitudes in surface soil moisture)**. Surface soil moisture is chosen as the scaling variable because*  *the GPP amplitude is strongly linearly related to the surface soil moisture amplitude (see analysis later in section). ESA CCI soil moisture is used as the benchmark observation. This scaling accounts for the possibility that individual models may under- or overestimate the regional surface soil moisture perturbation at the regional scale following wet events. Whilst a larger surface soil moisture perturbation is expected to lead to a larger-amplitude response in GPP, the scaled data allows us to assess the sensitivity of the GPP response to surface soil moisture by adjusting to identical surface soil moisture perturbations for each model.*

*L216: Readers may face difficulties locating relevant analysis later in this section. Maybe it is better to write: "see Figure 4 and analysis later in the section". Or a clearer way of locating the relevant analysis.*

We will change the text at line 216 to "see Figure 4 and accompanying analysis later in the section".

*Figure 3: Perhaps there could be an introduction that using the ratio as the radius is a method of normalization.*

We will adjust from line 235 as follows:

Figure 3 shows the correlation between regional responses for each possible model-observation pair, in addition to the  inter-regional standard deviation normalised by the observed standard deviation ($\sigma_{model}/\sigma_{obs}$) and the root mean squared error normalised by the observed standard deviation.

*3 Technical Corrections*

*Figure 3: "gpp" should be "GPP".*

*Figure 4: "vpd" should be "VPD".*

*Figure 1A: "gpp" and "vpd" should be capitalized.*

We will capitalise these as suggested in the revised manuscript.

**Reviewer 2**

*The manuscript titled "Contrasting responses of vegetation productivity to intraseasonal rainfall in Earth System Models" by Bethan L. Harris and other co-authors mainly evaluate vegetation productivity's response to water availability changes at the intraseasonal scale in in five CMIP6 Earth System Models (ESMs) and also compare the responses with a variety of observation-based products. The results mainly suggests that the models correctly capture that larger increases in GPP at the regional scale are associated with larger increases in surface soil moisture and larger decreases in atmospheric vapour pressure deficit (VPD). However, the sensitivity of the GPP response to these drivers varies between models.*

*Overall, the results are not new to the reader, and there are also many irregular expressions in terms of the figures and contents. The authors only used five ESMs to study the relationship between productivity and precipitation, and results suggest positive relationship between the two factors. I recommend the authors to dive into the models (or read more relevant model papers) to demonstrate the equations that used in the coupled land surface models to tell the reasons of the different responses between models.*

We would like to take this opportunity to clarify what we believe to be the novel results of this paper. As noted in lines 77–84 and highlighted by Reviewer 1, previous studies evaluating GPP in global models and across different observational products have investigated its behaviour at interannual timescales, whereas we focus on the intraseasonal timescale, which is also of known importance to vegetation variability (lines 85–86). Intraseasonal vegetation responses are important for determining its annual mean behaviour (Wu et al., 2021; line 361), so a good representation of the links between environmental drivers and GPP at this timescale is necessary to properly capture the longer-term behaviour of the climate system. We are not aware of any previous studies that have evaluated the GPP variability in CMIP6 models against observations at the intraseasonal timescale.

Our results concerning the sensitivity of the models provide useful new findings, particularly in the context of the observational evaluation we have carried out. For example, we find that NorESM has an unusually weak sensitivity to intraseasonal VPD perturbations compared to the other models studied. Without the observational evidence from our analysis, we would not know that the sensitivity is *too* weak—it could have been the case that all the other models had a sensitivity that is too strong. This is key information for future model development. By using multiple observational products, we have sought to provide the most robust evaluation possible for the models, and we are not aware of another study that has so thoroughly evaluated this sensitivity, particularly at the intraseasonal timescale.

To ensure that the novel aspects of the paper are clearer to the reader, we will highlight the intraseasonal timescale further in the abstract (line 2):

*We evaluate this response at the intraseasonal timescale in five CMIP6 ESMs by analysing changes in Gross Primary Productivity (GPP) after intraseasonal rainfall events* with a timescale of approximately 25 days.  We compare these responses to  those *found in a range of observation-based products.*

Our original manuscript did not include details on the importance of analysing the intraseasonal timescale rather than just focusing on interannual variability until late in the introduction (lines 77–90). To emphasise this novelty further we will summarise the argument in the abstract:

*Correctly representing the response of vegetation productivity to water availability in Earth System Models (ESMs) is essential for accurately modelling the terrestrial carbon cycle and the evolution of the climate system. Previous studies evaluating Gross Primary Productivity (GPP) in ESMs have focused on annual mean GPP and interannual variability, but physical processes at shorter timescales are important for determining vegetation-climate coupling. We evaluate GPP  responses at the intraseasonal timescale in five CMIP6 ESMs by analysing changes in  after intraseasonal rainfall events with a timescale of approximately 25 days and comparing to the responses found in a range of observation-based products.*

We only included five ESMs in the study because all the other models from the CMIP6 *esm-isv* experiment provided their GPP output data at monthly resolution, which is not sufficient for investigating the intraseasonal timescales that we have focused on (see lines 107-109). We will highlight in the discussion that it would be useful for more models to prioritise sub-monthly GPP output data for such experiments by adding the following at line 358:

*We therefore emphasise the usefulness of GPP data at the daily timescale for probing process-based diagnostics of climate-carbon cycle coupling. Many models only make GPP data from CMIP experiments publicly available at monthly resolution. Future prioritisation of sub-monthly vegetation data would aid the investigation of process-oriented diagnostics that can help us understand land-atmosphere coupling in these models.*

We agree that digging deeper into the precise aspects of model configuration that cause the differences in GPP response is extremely important, and this is exactly the kind of work that we hope to motivate with this paper. However, scrutinising the models' equations would not by itself be sufficient to explain all aspects of the differences in GPP responses. The response is the result of the complex interaction of many factors including details of prescribed soil and vegetation parameters as well as the local climate, and the balance of environmental controls on GPP can only be uncovered using output data from model runs themselves. The suggested work therefore requires running experiments with the models, which is outside the scope of this current manuscript. We have planned future work comparing the intraseasonal GPP responses between versions of JULES (the land surface model used in UKESM) with alternative soil moisture stress parameters and stomatal conductance models. This will establish how these particular changes in configuration impact the coupling between water availability and vegetation productivity in JULES, in order to begin addressing some of these questions.

We believe that our paper provides a valuable contribution by identifying that differences in GPP responses between models can be found at the intraseasonal timescale and by providing an observational evaluation against a wide range of products. We provide the metrics for future studies to perform more detailed model-specific analysis and cross-model comparisons focusing on different parameterisation approaches.

*Please check that the abbreviations used in all figures are uniform and standardized. The "GPP" should be capitalized in the figures. And it is suggested to use "VPD" instead of "vpd" throughout the figures to represent vapor pressure deficit. You should use "Observation" rather than "obs".*

We will capitalise GPP and VPD as suggested in the revised figures. We will write out "Observation", or where spatial constraints in the figures make this difficult we will define "obs" to mean observations in the figure caption.

*The legend in Figure 2 (b, c, d) should include an explanation regarding the color of the 1:1 line. And it is suggested to include the fitting equation for each model in the figure.*

We will add an annotation to the appropriate legends to explain the 1:1 line. Whilst the linear fits are useful for guiding the eye in these plots, we do not consider the precise parameters of the fitting equations to be relevant to the discussion or results and therefore feel that including them would only make the figure more cluttered and difficult to interpret. Later in the manuscript, where the sensitivity parameter becomes important, we will add an additional figure to ensure that all the relevant properties are clearly displayed for the reader (see response to later comment).

*In most of the figures, there is r rather than r2, why?*

We consider the Taylor diagrams in Figure 3 to be the most effective way of displaying the model-observation comparison statistics, and Taylor diagrams require r to be the azimuthal coordinate (Taylor et al., 2001). We therefore used r throughout the figures to maintain consistency (for example, the r values in Figure 2 directly correspond to points on the Figure 3 Taylor diagrams). We note from a later comment that our use of $r^2$ values in the text caused confusion since only r is visible in the figures. We will therefore alter this part of the text to ensure that the source of all quoted values is clear (see response to later comment).

*Line 265 to 272 is poor logic flow, please give the explanations.*

Thank you for highlighting this issue. We will rewrite this section as follows to clarify the argument.

*We now consider  the possible causes of such large differences between models. Different inter-model GPP responses may arise due to differing land cover maps (e.g. if forests show a weaker response to rainfall events than other land covers, then a model with more extensive forest cover will produce a weaker globally-composited GPP response). Alternatively, differences in the GPP response could be due to  variations in the models' representation of processes linking water availability and vegetation productivity, so that changes in GPP differ between models even when identical land covers are being considered.*

*To assess whether the first possibility is solely responsible for the differing GPP responses seen in Figure 1c, we compare regional land cover fractions between the models (Appendix A). We find that inter-model differences in the amplitude and lag of GPP responses can be seen even in regions where the models have similar land cover. Therefore, land cover differences cannot by themselves explain the discrepancies in the GPP response, so we further investigate the differences in water-carbon coupling processes. Two  important processes  that have known uncertainty in their representation in ESMs are the control of soil moisture stress on GPP and the response of stomatal conductance to changes in vapour pressure deficit (VPD). We therefore investigate the regional relationships between the GPP response and the perturbations in surface soil moisture (SSM) and near-surface  following the wet events.*

*How to see the negative correlation between SSM and VPD in Lines 278 to 279? Please give the explanations.*

We did not intend for the reader to be able to see this from the figures, as it is a commonly-known relationship and not directly relevant to the results. We will add "(not shown)" on line 279 to make this clear.

We will add the relevant explanation and equations for this at line 111–112:

*We compute daily near-surface VPD from near-surface air temperature (tas) and relative humidity (hurs). This is done by first computing the saturation vapour pressure $e_s$ according to Tetens' formula (Bolton, 1980):*

$$e_s = \; 6.112 \exp\left(\frac{17.67 \times tas}{tas + 243.5}\right),$$

*where tas is expressed in degrees Celsius. The vapour pressure deficit is then*

$$\text{VPD} = e_s \left(1 - \frac{hurs}{100}\right).$$

*We have not seen any figure annotations starting from Line 265, and the findings does not correspond to the figures. For example, we could not identify any evidence in Figure 4 that supports your findings in Line 289 to 290. In Line 291 to 299, the article mentions a comparison of slopes. However, we couldn't find any indication of the slope value of the fitted curve in Figure 4, which should be marked in the Figure 4.*

Apologies that this was not clear. The findings detailed in Lines 289–290 are based on the $r^2$ values of the linear fits, which correspond to the r values displayed in Figure 4. We have explained our preference for using r in the figures earlier in this response, but we will change the text at line 289 to ensure it is obvious where the values are coming from:

*For example, in CNRM-ESM2-1, the correlation between the peak amplitude of the SSM perturbations and the peak amplitude of the GPP response is r=0.95, meaning that the magnitude of the SSM perturbations can explain 90% of the variance in the GPP peak amplitude between regions ($r^2$=0.90), in CNRM-ESM2-1, whereas in UKESM1-0-LL it only explains 57% of the variance (r=0.76, $r^2$=0.57).*

We agree that the values for sensitivity need to be displayed for the reader. Since the sensitivity analysis is so important to our results, rather than just listing slope values in Figure 4 we have decided to dedicate a separate new Figure 5 to displaying the sensitivities. This allows us to also include uncertainty information in a way that the reader can easily compare between models and observational products. The new figure is included below as Figure R4.

The new figure will be introduced at line 294:

*We test the significance of these differences in  sensitivity among all the models and all the observational GPP products. This is done using estimated marginal means of linear trends and correcting for multiple testing using the Tukey method. Figure 5 shows the resulting 95% confidence intervals for the sensitivity of the GPP responses to the driving perturbations in SSM and VPD.*

[Figure]

*Figure R4: Sensitivity of regional GPP responses to intraseasonal wet events compared to the driving perturbations in surface soil moisture (a, c, e) and VPD (b, d, f). Errorbars denote 95% confidence intervals. The sensitivity of the observational GPP products is measured with respect to ESA CCI for surface soil moisture and ERA5 for VPD.*

While producing this new figure, we identified an error in the computation of the sensitivities. This affects some of the conclusions, but we still find significant differences between models and observations once the computation is carried out correctly. None of the original figures were affected by this error. The text will be updated to reflect the correct conclusions as follows.

Line 296:

*This analysis shows that for the relationship between the peak amplitudes of  VPD and GPP, the sensitivity of NorESM is significantly different ($p <$ 0.1) to the sensitivity of all other models and all observational GPP products except SIF-GOME2-JJ. These differences are also all significant at p<0.05 other than NorESM-VODCA2GPP.  Therefore, we conclude that the amplitude of GPP responses to wet events in NorESM is not sufficiently sensitive to the changes in vapour pressure deficit.*

*The peak amplitude of GPP responses is also less sensitive to SSM perturbations in NorESM than in the other models, showing a significantly (p<0.05) different sensitivity to BCC-CSM, CNRM-ESM and UKESM. However, of the 7 observational products, the SSM sensitivity in NorESM is only significantly different to FLUXCOM-ERA5, so there is insufficient evidence here to state that it is unrealistic compared to the range of observations.  In contrast, the peak amplitude of GPP responses in BCC-CSM shows excessive sensitivity to SSM perturbations, with significant (p<0.05) differences to MODIS Terra, VPM, SIF-GOME2-JJ and VODCA2GPP (and p<0.1 for the difference with FLUXCOM-CRUJRAv1). It additionally shows a significantly higher sensitivity than two of the other models (NorESM and ACCESS-ESM).*

Line 340:

*In particular, GPP responses in BCC-CSM show excessive sensitivity to intraseasonal surface soil moisture perturbations, while NorESM is not sufficiently sensitive to vapour pressure deficit perturbations. This aligns with the findings of Anav et al. (2015) that NorESM has lower seasonal and interannual variability than other models.*

Line 10:

*The GPP in NorESM is insufficiently sensitive to vapour pressure deficit perturbations  compared to all models and  6 out of 7 observational GPP products tested.*

*Line 105, from 1850-2014 should be from \*\* to \*\* or during 1850-2014*

We will change this to "from 1850 to 2014".

*Line 359, the vertical profile of root water uptake may also play a role. We all know that, but how? Give some evidence in terms of model equations or parameters?*

We have deliberately not presented specific model equations in this section as, without data from model sensitivity experiments, we would only be able to speculate as to how a particular aspect of configuration is actually impacting the GPP responses we see. With so many complex interactions between processes at play, we do not believe that quoting any model equations or parameters would provide conclusive evidence for the response differences. However, we did want to highlight the areas of the models that it would be useful for future development efforts to specifically investigate in order to generate this evidence.

*Line 355, "Further work is therefore needed to understand why different methods for deriving global GPP products result in different relationships with water availability, quantify the uncertainty in these products, and ultimately to obtain observations that will reduce our uncertainty in the response of GPP to intraseasonal rainfall events" Based on my understanding, this is what this work really need to answer, or I can not find new things in this work.*

As discussed in response to the opening comments of the review, the novel aspects of the paper are the focus on the GPP response at the intraseasonal timescale rather than interannual, and the results uncovering sensitivity differences across models and a wide range of observational products. We do not consider it to be within the scope of this paper to assess the precise reasons for differences between the observational products, which are produced using complex and diverse methods, but we believe that uncovering the differences is a valuable contribution in itself. Prior to performing our analyses, we did not know how different the intraseasonal GPP responses would be between observational products, and indeed we were surprised by how marked the discrepancies are. Our work is a crucial step in motivating future efforts towards understanding these observational inter-product differences and improving our quantification of carbon-water cycle coupling.

**References**

Bolton, D. (1980). The Computation of Equivalent Potential Temperature. Monthly Weather Review, 108(7), 1046–1053.
https://doi.org/10.1175/1520-0493(1980)108%3C1046:TCOEPT%3E2.0.CO;2

Iturbide, M., Gutiérrez, J. M., Alves, L. M., Bedia, J., Cerezo-Mota, R., Cimadevilla, E., Cofiño, A. S., Luca, A. di, Faria, S. H., Gorodetskaya, I. v., Hauser, M., Herrera, S., Hennessy, K., Hewitt, H. T., Jones, R. G., Krakovska, S., Manzanas, R., Martínez-Castro, D., Narisma, G. T., … Vera, C. S. (2020). An update of IPCC climate reference regions for subcontinental analysis of climate model data: definition and aggregated datasets. Earth System Science Data, 12(4), 2959–2970.
https://doi.org/10.5194/ESSD-12-2959-2020

Taylor, K. E. (2001). Summarizing multiple aspects of model performance in a single diagram. Journal of Geophysical Research: Atmospheres, 106(D7), 7183–7192.
https://doi.org/10.1029/2000JD900719

Wu, M., Vico, G., Manzoni, S., Cai, Z., Bassiouni, M., Tian, F., Zhang, J., Ye, K., & Messori, G. (2021). Early Growing Season Anomalies in Vegetation Activity Determine the Large-Scale Climate-Vegetation Coupling in Europe. Journal of Geophysical Research: Biogeosciences, 126(5), e2020JG006167.
https://doi.org/10.1029/2020JG006167

---

## Referee Report (RR1)

**Referee Report**

The author provided detailed responses to my comments in the initial review round. This enhanced the clarity and accuracy of the article, leading me to recommend its acceptance.

In the initial review, I raised three critical points concerning spatial resolution, sample size, and response variables. The author not only responded adequately to these issues but also supplemented experimental data to address my questions. Firstly, the author accurately defined the research subject to 25-day precipitation events, thus avoiding potential reader misinterpretations. Secondly, through the sensitivity analysis of data spatial resolution, it was demonstrated that the experimental results remained stable within the range of 0.1 to 1 degree, corresponding to the spatial scale of intra-seasonal precipitation events. The author also provided information on the sample size for computing regional composites, indicating that the minimum sample size exceeded 500 precipitation events, ensuring a high level of confidence. It is noteworthy that the author adjusted the research region to 60°S-60°N, after identifying significant differences in sample sizes of various datasets in high-latitude regions, effectively mitigating the uncertainty associated with meteorological data in these regions. Concerning the third issue, regarding the GPP variation range or its sensitivity to extreme precipitation, the author chose to focus on sensitivity, which is acceptable.

I greatly appreciate the author's clear and comprehensible responses to comments about methods. For instance, regarding the comment on [line 154 (original script)], the author detailed the process of utilizing the Lanczos filtering method to identify precipitation events, making it easier for readers to comprehend. Regarding the comment on [line 213 (original script)], the author explained in detail the reason behind choosing soil moisture for recalibration, significantly enhancing the logical coherence of the article.

A slight drawback was the response to comment about [line 206 (original script)], where the author explained that the 40-60 day window was set to avoid the influence of repeated precipitation events. While this slightly differs from my original intention, which emphasized that vegetation responses continue after this period and may not be entirely assessed, it does not affect the core content of the article and can be further explored in future work.

In conclusion, following the revisions made by the author to the initial draft, the overall quality of the article has significantly improved. I recommend accepting this article.